# Genotype-Specific HPV mRNA Triage Improves CIN2+ Detection Efficiency Compared to Cytology: A Population-Based Study of HPV DNA-Positive Women

**DOI:** 10.3390/pathogens14080749

**Published:** 2025-07-30

**Authors:** S. Sørbye, B. M. Falang, M. Antonsen, E. Mortensen

**Affiliations:** 1Department of Clinical Pathology, University Hospital of North Norway, 9006 Tromsø, Norway; mona.antonsen@unn.no (M.A.); elin.mortensen@unn.no (E.M.); 2PreTect AS, 3490 Klokkarstua, Norway; bente.falang@pretect.no; 3Department of Medical Biology, Faculty of Health Sciences, University of Tromsø, 9006 Tromsø, Norway

**Keywords:** HPV screening, HPV mRNA testing, E6/E7 oncogene expression, cervical intraepithelial neoplasia (CIN2+), molecular triage, self-sampling, colposcopy referral, cervical cancer prevention, genotype-specific risk stratification, liquid-based cytology

## Abstract

Background: Effective triage of women testing positive for high-risk HPV DNA is essential to reduce unnecessary colposcopies while preserving cancer prevention. Cytology, the current standard, has limited specificity and reproducibility. The genotype-specific 7-type HPV E6/E7 mRNA test (PreTect HPV-Proofer’7), targeting HPV types 16, 18, 31, 33, 45, 52, and 58, detects transcriptionally active infections and may enhance risk stratification. Methods: Between 2019 and 2023, 34,721 women aged 25–69 underwent primary HPV DNA screening with the Cobas 4800 assay at the University Hospital of North Norway, within the national screening program. Of these, 1896 HPV DNA-positive women were triaged with liquid-based cytology with atypical squamous cells of undetermined significance or worse (≥ASC-US) and the 7-type HPV mRNA test. Histological outcomes were followed through October 2024. Diagnostic performance for CIN2+ was evaluated overall and by genotype. Results: CIN2+ prevalence was 13.3%. The mRNA test reduced test positivity from 50.3% to 33.4% while maintaining comparable sensitivity (70.6% vs. 72.2%) and improving specificity (72.3% vs. 53.0%) and PPV (28.1% vs. 19.1%). Genotype-specific PPVs were highest for HPV16 mRNA (47.7%), followed by HPV33 (39.2%) and HPV31 (32.2%), all exceeding corresponding DNA-based estimates. Conclusion: Genotype-specific HPV mRNA triage offers superior risk discrimination compared to cytology, supporting more targeted, efficient, and accessible cervical cancer screening.

## 1. Introduction

Cervical cancer is a preventable disease, with persistent infection by oncogenic human papillomavirus (HPV) types recognized as its necessary cause [1]. As a result, HPV testing has become central to cervical cancer prevention strategies, gradually replacing cytology in primary screening programs due to its superior sensitivity in detecting high-grade cervical intraepithelial neoplasia (CIN2+) and cancer [2]. However, the reduced specificity of HPV DNA-based tests, particularly among younger women with transient infections, remains a clinical challenge, often leading to unnecessary colposcopy referrals and overtreatment [3].

To address this, several triage strategies have been proposed for HPV DNA-positive women. Cytology remains the current standard, but its inherent subjectivity and limited reproducibility constrain its effectiveness. Partial genotyping offers some improvement by identifying high-risk types, yet it does not capture the transcriptional activity that drives disease progression. Additional molecular-based triage approaches, including dual-stained cytology (p16/Ki-67) and DNA methylation markers, have shown promise in improving objectivity and sensitivity [4,5]. However, these methods often rely on cytological infrastructure, have variable specificity in real-world settings [4,6], and are not validated for self-collected samples, limiting their scalability in population-based programs. Thus, there remains a need for validated triage tools that combine high specificity with operational flexibility.

As self-sampling gains traction in organized screening programs, triage modalities that can be applied to self-collected samples are increasingly important [7]. This has spurred growing interest in HPV mRNA testing, which detects active transcription of the E6 and E7 viral oncogenes—key drivers of malignant transformation. Unlike HPV DNA detection, which reflects exposure, mRNA testing identifies transcriptionally active, clinically relevant infections and may offer improved risk stratification [8,9,10,11,12].

The PreTect HPV-Proofer’7 is a genotype-specific mRNA test targeting E6/E7 transcripts from the seven most oncogenic HPV types (16, 18, 31, 33, 45, 52, and 58), all of which are included in the nonvalent HPV vaccine and account for the majority of cervical cancer cases worldwide. Previous studies have demonstrated that HPV mRNA testing provides higher specificity and positive predictive value (PPV) than cytology or HPV DNA genotyping in triage settings [13,14,15].

Building on prior data from the initial implementation phase, this study provides a substantially expanded analysis based on a doubled cohort size (1896 women) and five years of real-world triage data. In addition to updated performance metrics with extended follow-up through October 2024, we present detailed genotype-specific risk stratification analyses not previously reported. These results offer new evidence supporting the clinical utility and implementation feasibility of the 7-type HPV mRNA assay in population-based cervical cancer screening. The objective of this study is to evaluate the clinical performance of the genotype-specific 7-type HPV mRNA assay as a triage tool for HPV DNA-positive women, comparing its diagnostic accuracy and risk stratification ability to cytology.

## 2. Materials and Methods

### 2.1. Study Design and Population

This quality-assurance cohort study was embedded in the Norwegian Cervical Cancer Screening Programme (NCCSP) and conducted at the Department of Clinical Pathology, University Hospital of North Norway (UNN), Tromsø. The study design, inclusion criteria, and screening procedures have been described in detail in a previous publication covering the initial implementation period from 2019 to 2021 [14]. The present analysis represents an updated and expanded evaluation, extending the study period through 31 December 2023, with follow-up for histologically confirmed outcomes through 31 October 2024. Eligibility criteria and exclusion parameters remained consistent with those used in the earlier cohort.

The study population consisted of women undergoing routine primary HPV screening. Women with a history of cervical cancer, prior treatment for CIN2+ within the last 10 years, ongoing follow-up for previous abnormal findings, or known immunosuppression were not included in the primary screening cohort. Completeness of follow-up was ensured through the national call/recall system and use of unique personal identifiers, which enable tracking of all cervical screening and follow-up data across laboratories and healthcare providers nationwide.

#### 2.1.1. Primary HPV DNA Screening

Samples were collected in PreservCyt^®^ solution (ThinPrep^®^, Hologic Inc., Marlborough, MA, USA) and analyzed using the Cobas 4800 HPV DNA assay (Roche Diagnostics, Mannheim, Germany). This assay individually detects HPV genotypes 16 and 18 and reports a pooled result for 12 other high-risk (HR) HPV types.

#### 2.1.2. Triage Procedures

Women who tested positive for high-risk HPV DNA underwent two parallel triage assessments, both performed using residual material from the same PreservCyt^®^ sample:

Liquid-Based Cytology (LBC): Cytological evaluation was conducted according to the 2014 Bethesda system. A result of atypical squamous cells of undetermined significance or worse (≥ASC-US) was considered positive;

7-Type HPV E6/E7 mRNA Assay: The PreTect HPV-Proofer’7 assay (PreTect AS, Klokkarstua, Norway) detects transcriptionally active infections with HPV genotypes 16, 18, 31, 33, 45, 52, and 58, providing genotype-specific results for each target individually rather than as a pooled result. The assay includes an intrinsic sample adequacy control (ISC) targeting mRNA from a human housekeeping gene (GAPDH), ensuring RNA integrity and sample validity. RNA was extracted from 1 mL of residual liquid-based cytology (LBC) specimen using the PreTect X protocol, with elution in 80 µL. Amplification was performed using nucleic acid sequence-based amplification (NASBA), an isothermal RNA amplification method conducted at 41 °C, according to the manufacturer’s instructions. All testing was completed within six weeks of collection to minimize RNA degradation.

Data interpretation was performed using PreTect Analysis Software (PAS), version 6.12 (PreTect AS), a dedicated software platform that verifies run validity by confirming that all assay control criteria are met, including positive detection of all genotype-specific positive controls, absence of signal in negative controls, and successful intrinsic sample control (ISC) detection in each individual sample. For each fluorescence dye/well, PAS computes a signal-to-cutoff (S/CO) ratio based on the maximum relative fluorescence unit (RFU) and the calculated baseline RFU. Results are classified using predefined thresholds: S/CO < 1.3 is considered negative (mRNA not detected); S/CO ≥ 1.5 is considered positive (mRNA detected); and S/CO values between 1.3 and 1.5 are interpreted as indeterminate. In accordance with the assay operator manual, all indeterminate results were retested to yield a conclusive result.

Reproducibility of the PreTect HPV-Proofer’7 assay has been consistently demonstrated through internal quality control procedures implemented during routine clinical testing. While the present study focuses on clinical performance, the assay has undergone comprehensive analytical validation in accordance with United States CLIA standards. This validation includes assessment of key performance characteristics such as accuracy, precision, analytical sensitivity and specificity. The assay is currently implemented as a laboratory-developed test (LDT) within a CAP-accredited clinical laboratory in Atlanta, Georgia. The assay has also been validated for use on self-collected vaginal specimens, supporting its applicability in decentralized screening settings [16,17].

### 2.2. Follow-Up and Outcome Ascertainment

Clinical management followed the national guidelines in effect at the time of testing. HPV mRNA testing was performed retrospectively and did not influence clinical management or timing of diagnosis, which was based solely on cytology and HPV DNA results according to national guidelines. Where indicated, colposcopy-directed biopsies or systematic four-quadrant biopsies of the transformation zone were performed. The primary study endpoint was histologically confirmed cervical intraepithelial neoplasia grade 2 or worse (CIN2+), coded according to World Health Organization (WHO) criteria. Pathologists evaluating histology were blinded to the mRNA test results. Outcome data were collected through 31 October 2024.

All high-grade cervical biopsies (CIN2+) were evaluated by two experienced pathologists. CIN grading was performed according to WHO criteria, with p16 immunostaining used in morphologically ambiguous cases to distinguish CIN1 from CIN2+. Digital pathology and machine learning-based quality assurance tools (including EagleEye AI) were used to support diagnostic consistency. In cases of discrepancy, the final diagnosis was determined by the evaluating pathologist.

### 2.3. Statistical Analysis

Sensitivity, specificity, positive predictive value (PPV), and negative predictive value (NPV) for CIN2+ detection were calculated for each triage modality (LBC ≥ ASC-US and the 7-type HPV mRNA test). Exact 95% confidence intervals (CIs) were calculated using the Clopper–Pearson method. Paired comparisons between test modalities were conducted using McNemar’s test, with *p*-values < 0.05 considered statistically significant. All analyses were performed using IBM SPSS Statistics for Windows, Version 29.0 (IBM Corp., Armonk, NY, USA, 2022).

### 2.4. Ethical Approval

This study was approved by the Regional Committee for Medical and Health Research Ethics, North Norway, as a program evaluation project (REK Nord 203384). In accordance with Norwegian regulations, individual informed consent was not required for quality-assurance studies based on de-identified registry data.

## 3. Results

### 3.1. Primary Screening Outcomes and Triage Cohort Formation

Among 34,721 women screened with the Cobas 4800 HPV DNA assay between January 2019 and December 2023, 1944 (5.6%) were HPV DNA-positive. After exclusion of 48 samples with insufficient volume or invalid mRNA internal control, 1896 women (97.5%) comprised the analytic triage cohort, doubling the size of our previous interim report (n = 962) and extending follow-up to 31 October 2024 (Figure 1).

### 3.2. Comparative Triage Positivity Rates and Impact on Referral Burden

Among the 1896 HPV DNA-positive women, 50.3% (954/1896) had abnormal cytology results (≥ASC-US), while 49.7% (942/1896) were cytology-negative. In contrast, the 7-type HPV mRNA assay was positive in 33.4% (634/1896) and negative in 66.6% (1262/1896) of cases. This reflects a 34% relative reduction in test positivity compared to cytology. Notably, 320 women had abnormal cytology but tested negative on the mRNA assay; these women would not have been referred for colposcopy if mRNA triage had been used instead of cytology, indicating improved specificity and reduced over-referral with the mRNA test (Figure 1).

### 3.3. CIN2+ Detection Rates and Risk Stratification by Triage Modality

During follow-up, 252 women were diagnosed with CIN2+, yielding a prevalence of 13.3% (252/1896), virtually identical to the 13.9% observed in the earlier analysis. Among women with abnormal cytology (≥ASC-US), 19.1% (182/954) had CIN2+, compared to 7.4% (70/942) with normal cytology. For the 7-type HPV mRNA test, 28.1% (178/634) of mRNA-positive women had CIN2+, whereas only 5.9% (74/1262) of mRNA-negative women had CIN2+, demonstrating the improved risk discrimination provided by genotype-specific mRNA triage (Figure 2).

### 3.4. Comparative Diagnostic Accuracy of Cytology and HPV mRNA Triage for CIN2+ Detection

The diagnostic accuracy of the two triage strategies for detecting CIN2+ is summarized in Table 1 and Figure 3. Sensitivity was comparable between cytology (72.2%; 95% CI: 66.3–77.5) and the 7-type HPV mRNA test (70.6%; 95% CI: 64.6–75.9), with a minimal absolute difference of −1.6 percentage points. McNemar’s test showed no statistically significant difference in sensitivity between the two methods (*p* = 0.13).

However, the mRNA test demonstrated substantially higher specificity (72.3% vs. 53.0%), representing a 19.3 percentage point increase and a ~1.4-fold improvement relative to cytology. This difference in specificity was statistically significant (McNemar’s test, *p* < 0.001). Positive predictive value (PPV) increased from 19.1% with cytology to 28.1% with mRNA (Δ = +9.0 pp), a 47% relative improvement. Negative predictive value (NPV) was high for both tests, with a slight advantage for mRNA (94.6% vs. 93.0%). These findings highlight the superior risk discrimination achieved with genotype-specific mRNA testing. While statistical comparisons were not included in the table, the clinical implications of these differences—particularly the improved specificity and PPV—support the utility of HPV mRNA triage in reducing unnecessary colposcopies and focusing follow-up on women at highest risk.

### 3.5. Colposcopy Efficiency: Procedures Required per CIN2+ Case Detected

Applying cytology ≥ ASC-US would lead to 5.2 colposcopies per CIN2+ detected (954/182), whereas the mRNA strategy would require 3.6 colposcopies per CIN2+ (634/178), reflecting a 31% reduction in procedures for comparable disease yield.

### 3.6. Genotype-Specific Predictive Values

The 7-type HPV mRNA assay provided improved risk stratification compared to HPV DNA genotyping. The positive predictive value (PPV) for CIN2+ was substantially higher for HPV16 mRNA (47.7%) compared to HPV16 DNA (29.8%), and for HPV18 mRNA (30.2%) versus HPV18 DNA (22.0%). Among women with mRNA-positive results for other genotypes (31, 33, 45, 52, 58), the overall PPV was 22.0%, compared to 10.2% for non-16/18 HPV DNA types. Genotype-specific PPVs for the mRNA assay were highest for HPV33 (39.2%), HPV31 (32.2%), HPV52 (24.1%), and HPV45 (12.3%), underscoring the ability of genotype-specific mRNA testing to differentiate clinically meaningful high-grade lesions from transient ones (Figure 4). The overall PPV across all mRNA-positive samples was 28.1%, reinforcing its utility in triaging HPV DNA-positive women.

In this expanded cohort, 7-type HPV E6/E7 mRNA triage preserved sensitivity, increased specificity by nearly 20 percentage points, and reduced the number of colposcopies required to detect one CIN2+ case from 5.2 to 3.6, confirming and strengthening the advantages previously observed in the interim analysis. This enhanced genotype-specific stratification may be particularly valuable in partially vaccinated populations, where the prevalence of certain HPV types is reduced and residual disease risk is increasingly concentrated in a smaller subset of high-risk genotypes.

### 3.7. Refining Risk Stratification in HPV16/18 DNA-Positive Women and Other Genotype Subgroups

Stratification of HPV DNA-positive women by mRNA status revealed marked differences in CIN2+ risk within genotype groups. Among women positive for HPV DNA type 16, the overall CIN2+ rate was 29.8%, but this increased to 47.7% in those also positive for HPV16 mRNA and dropped to 9.7% in those mRNA-negative (Figure 5). Similarly, for women with HPV18 DNA, the overall CIN2+ rate was 22.2%, with 30.2% in the mRNA-positive group and 10.5% in the mRNA-negative group (Figure 6).

For non-16/18 HPV genotypes, the overall CIN2+ rate was 10.2% among women testing positive for any of the 12 high-risk HPV types included in the pooled DNA result. However, when stratified by genotype-specific mRNA detection using the PreTect HPV-Proofer’7 assay, analysis of the five non-16/18 genotypes included (HPV31, 33, 45, 52, and 58) showed that the CIN2+ risk increased to 22.0% among mRNA-positive women and was only 5.6% among mRNA-negative cases (Figure 7). Among these, HPV33 mRNA demonstrated the highest genotype-specific PPV (39.2%), followed by HPV31 (32.2%) and HPV52 (24.1%) (Figure 4), highlighting the assay’s ability to discriminate risk not only for HPV16 and 18 but also for other oncogenic genotypes. These results reinforce the clinical utility of genotype-specific HPV mRNA detection in accurately stratifying CIN2+ risk across all high-risk types included in the assay.

## 4. Discussion

### 4.1. Clinical Utility of Genotype-Specific HPV mRNA Triage

This study demonstrates the clinical utility and substantial advantages of employing the 7-type HPV mRNA E6/E7 assay (PreTect HPV-Proofer’7) for triage of HPV DNA-positive women within a population-based cervical cancer screening program. Our findings confirm that this genotype-specific mRNA test achieves comparable sensitivity to liquid-based cytology while substantially improving specificity and positive predictive value (PPV), thereby addressing a key limitation of HPV DNA-based screening, its relatively low specificity and the consequent risk of overtreatment.

In our cohort, test positivity was substantially lower with HPV mRNA testing (33.4%) compared to cytology (50.3%), resulting in fewer women being referred for colposcopy without compromising sensitivity (70.6% vs. 72.2%). This reduction in over-referral aligns with previous reports evaluating HPV mRNA in both primary and delayed triage settings [14,18,19].

By detecting transcriptionally active infections, the mRNA assay offers superior risk discrimination and more effectively identifies women with clinically significant infections while avoiding unnecessary procedures.

In Norway, only p16-positive CIN2 lesions are considered clinically significant and are classified as CIN2. Ambiguous or p16-negative lesions are downgraded to CIN1 or reactive changes. All CIN2+ outcomes in this study were confirmed by two independent pathologists and, where needed, supported by p16 staining and digital pathology tools to ensure reproducibility and clinical relevance.

### 4.2. Enhanced Stratification Within HPV DNA 16/18

Importantly, genotype-specific analysis revealed that the mRNA assay considerably improved PPVs for HPV16, HPV18, and other high-risk types, enabling more refined CIN2+ risk stratification among HPV DNA-positive women. This level of detail is particularly valuable in the context of HPV-vaccinated populations, where the prevalence of vaccine-covered genotypes is decreasing and residual disease risk is increasingly concentrated in fewer, non-vaccine HPV types. Moreover, the ability of the mRNA assay to differentiate transcriptionally active, and therefore clinically meaningful, infections is particularly relevant in younger women, who have the highest HPV prevalence and are most likely to be diagnosed with transient CIN2 lesions [6,20,21]. In this group, cytology or DNA-based screening may lead to unnecessary interventions for regressive lesions, resulting in overtreatment and psychological burden. By focusing triage on transcriptionally active infections with known oncogenic potential, genotype-specific mRNA testing may help reduce overtreatment in young women while preserving sensitivity for detecting lesions with true progression risk [22].

### 4.3. Negative Predictive Value and Long-Term Safety of mRNA Triage

The observed negative predictive value of the 7-type mRNA test in our study (94.6%) was comparable to that of cytology-based triage (93.0%), supporting the safety of managing mRNA-negative women within current surveillance intervals. Although the assay targets only a subset of the 14 high-risk HPV genotypes detected by primary screening, prior long-term follow-up studies have demonstrated that HPV mRNA-negative women have a persistently low risk of CIN3+ over time. In a Norwegian cohort study involving the earlier 5-type version of the HPV mRNA test, the 10-year cumulative incidence of CIN3+ among mRNA-negative women was just 1.1%, comparable to the risk observed with negative results from extended genotype DNA tests [23]. These findings reinforce the clinical reliability of a negative HPV mRNA test and support its use as a triage tool with similar follow-up safety expectations as cytology.

### 4.4. HPV mRNA Versus Other Molecular Triage Strategies

Several molecular triage strategies have been proposed as alternatives to cytology for the management of HPV DNA-positive women, including p16/Ki-67 dual-stained cytology (CINtec^®^ PLUS) and DNA methylation assays. These approaches aim to improve sensitivity and objectivity, particularly in women with equivocal cytology or non-16/18 high-risk HPV infections [24,25,26,27,28,29,30].

CINtec PLUS has shown increased sensitivity and negative predictive value (NPV) compared to cytology, especially when combined with HPV16/18 genotyping, as demonstrated in studies by McMenamin et al. and Øvestad et al. [29,30]. However, reported specificity values have varied. Øvestad et al. found that dual staining led to lower specificity than cytology, implying more false-positive referrals and a higher colposcopy burden. In contrast, McMenamin et al. reported that CINtec PLUS improved specificity over HPV testing in women with equivocal cytology and could reduce referral rates by up to 40%. These inconsistencies highlight that the clinical utility of CINtec PLUS depends significantly on population characteristics, cytology quality, and the operational implementation (standalone vs. with genotyping).

Our findings demonstrate that the 7-type HPV mRNA test achieves a comparable reduction in colposcopy referrals, with test positivity decreasing from 50.3% with cytology to 33.4%–a 34% relative reduction—while also providing substantially higher specificity (72.3% vs. 53.0%) and a 47% relative increase in positive predictive value (28.1% vs. 19.1%). Unlike dual staining, HPV mRNA testing directly detects transcriptionally active oncogenic infections, offering enhanced clinical precision. In addition, genotype-specific results provide more refined risk stratification without the need for additional platforms or cytotechnologist interpretation.

Importantly, HPV mRNA testing is compatible with self-collected vaginal specimens, unlike cytology-based or dual-stained approaches that require clinician-collected samples and microscopy infrastructure. This feature improves feasibility in decentralized settings, reduces logistical barriers to screening, and facilitates broader population reach—particularly in low-resource contexts or programs relying on self-sampling strategies.

Moreover, evidence from Carcea et al. supports the superior clinical relevance of E6/E7 mRNA detection. In a study comparing mRNA testing with HPV DNA and p16/Ki-67 immunostaining for detecting post-treatment CIN2+ recurrence, the mRNA assay achieved the highest sensitivity (100%) and specificity (96.9%), and was the only method significantly correlated with histological severity (ρ = 0.345, *p* = 0.006) [26].

Taken together, the clinical performance, operational flexibility, and self-sampling compatibility of the 7-type HPV mRNA assay position it as a highly promising molecular triage tool for organized cervical cancer screening programs.

While dual-stained cytology and methylation markers have shown promise as triage tools [27,28,29,30], some mRNA assays, such as the 14-type APTIMA test, have demonstrated lower specificity than cytology or dual-staining in head-to-head comparisons [31]. In contrast, the genotype-specific 7-type HPV mRNA assay evaluated in this study demonstrated significantly higher specificity than cytology, supporting its potential for reducing unnecessary colposcopy referrals.

### 4.5. Colposcopy Efficiency and Health System Impact

The reduction in the number of colposcopies required per CIN2+ case, from 5.2 with cytology to 3.6 with mRNA testing, illustrates both the clinical efficiency and potential cost-effectiveness of mRNA-based triage. These findings are consistent with results from the initial implementation phase (2019–2021), which reported similar colposcopy efficiency (5.2 vs. 3.4) based on a cohort of 962 women [14]. The current expanded analysis, based on a doubled sample size and extended follow-up through October 2024, reinforces the robustness of these estimates. This consistency across time and population size underscores the reliability of genotype-specific HPV mRNA testing in reducing unnecessary procedures and improving the overall benefit–harm balance of cervical cancer screening.

### 4.6. Alignment with National Data on HPV Genotype Risk Stratification

Our findings on genotype-specific risk stratification are broadly consistent with earlier population-based data from the Norwegian Cervical Cancer Screening Programme. In the study by Hashim et al. (2020), the estimated CIN3+ risks among HPV DNA-positive women with normal cytology were 19.9% for HPV16, 10.8% for HPV18, and 5.5% for other high-risk HPV types [32]. In our cohort, using a genotype-specific mRNA triage approach, we observed lower absolute risks within each group—9.7% for mRNA-negative HPV16, 10.5% for mRNA-negative HPV18, and 5.6% for mRNA-negative non-16/18 types—indicating effective de-escalation of risk among women with transcriptionally inactive infections. Conversely, CIN2+ risk among women who were mRNA-positive for HPV16 was 47.7%, compared to 29.8% for HPV16 DNA alone. These differences underscore the enhanced discriminatory capacity of HPV mRNA testing to differentiate clinically relevant infections from transient ones, especially within genotype categories. By more accurately identifying women at low versus high risk, genotype-specific mRNA testing may allow for safer postponement of colposcopy in mRNA-negative women while maintaining robust detection of high-grade lesions among those with transcriptionally active infections. This reinforces the added value of transcriptional activity detection over DNA presence alone in stratifying risk.

### 4.7. Study Limitations and Need for Further Validation

Several limitations should be acknowledged. First, the follow-up duration may be insufficient to fully capture the long-term safety and predictive value of mRNA-based triage. Second, the lack of stratification by HPV vaccination status is a limitation, especially in the context of evolving genotype prevalence in vaccinated cohorts. Although most women in this study were born before the national HPV vaccination program began in 2009—and were thus likely unvaccinated—the absence of individual-level vaccination data precluded definitive subgroup analysis. Third, while our findings support the safety of standard surveillance intervals for mRNA-negative women, it should be noted that the 7-type assay does not cover all 14 high-risk HPV types included in the primary DNA screening test. Although long-term data from a similar 5-type mRNA test indicate durable protection against CIN3+ [23], further real-world studies evaluating the full 7-type assay over extended follow-up are needed to confirm long-term safety in more diverse screening populations.

Few studies have evaluated the performance of HPV mRNA triage in vaccinated populations. As vaccination coverage continues to rise and genotype prevalence shifts, future research will be essential to assess the effectiveness and clinical utility of mRNA-based triage strategies in vaccinated cohorts.

### 4.8. Interpretation of Diagnostic Differences and Clinical Relevance

Differences in diagnostic performance between HPV mRNA triage and cytology should be interpreted primarily in terms of clinical relevance. Although the increase in specificity and negative predictive value observed for the HPV mRNA test is substantial, these absolute improvements—while modest—may translate into meaningful clinical benefits in large-scale screening programs by reducing unnecessary colposcopies. Moreover, the relative increase in positive predictive value (47%) with mRNA triage represents a clinically important gain, indicating more accurate identification of women at highest risk of CIN2+. The interpretation of such differences should therefore prioritize clinical impact rather than relying solely on statistical significance, particularly given the paired nature of the data and the large sample size involved.

In addition to diagnostic improvements, HPV mRNA triage may offer advantages in cost-effectiveness and implementation feasibility. Cytology is resource-intensive, subjective, and dependent on experienced personnel, which poses challenges in many countries, including Norway, where cervical samples are primarily collected by general practitioners and access to gynecologists is limited. While the HPV mRNA test involves a small additional cost, its ability to reduce unnecessary colposcopies and biopsies could offset this by decreasing the workload for specialist care and improving the efficiency of the screening program. Formal cost-effectiveness analyses should be conducted in future studies to quantify these potential benefits.

Although the sensitivity of the HPV mRNA assay was comparable to that of cytology, no screening test achieves perfect sensitivity. Nonetheless, clinical safety was not compromised in this study, as all women were managed according to HPV DNA and cytology results, regardless of mRNA status.

In a future setting where mRNA replaces cytology as the sole triage modality, the 7-type mRNA assay—demonstrating significantly higher specificity and statistically comparable sensitivity to cytology—may offer a safe and effective means to reduce unnecessary colposcopy referrals, provided that HPV DNA-positive, mRNA-negative women are managed with continued follow-up in accordance with clinical guidelines.

In summary, the 7-type HPV mRNA assay significantly improves specificity and PPV over cytology while maintaining high sensitivity. Its applicability to self-collected samples, capacity for genotype-specific risk discrimination, and ability to reduce unnecessary colposcopies support its broader implementation as a precision triage tool in HPV-based cervical cancer screening programs.

## 5. Conclusions

This study confirms that triage of HPV DNA-positive women using the 7-type HPV mRNA assay (PreTect HPV-Proofer’7) maintains diagnostic sensitivity while markedly improving specificity and positive predictive value compared to cytology-based triage. The use of genotype-specific mRNA detection substantially reduces unnecessary colposcopy referrals, mitigating overtreatment and patient burden. Moreover, validated compatibility with self-collected vaginal specimens supports broader implementation, particularly in screening programs incorporating self-sampling. This approach enables risk-adapted clinical management based on genotype-specific transcriptional activity, efficient allocation of clinical resources, and improved cost-effectiveness, key principles of precision medicine in cervical cancer prevention.

## Figures and Tables

**Figure 1 pathogens-14-00749-f001:**
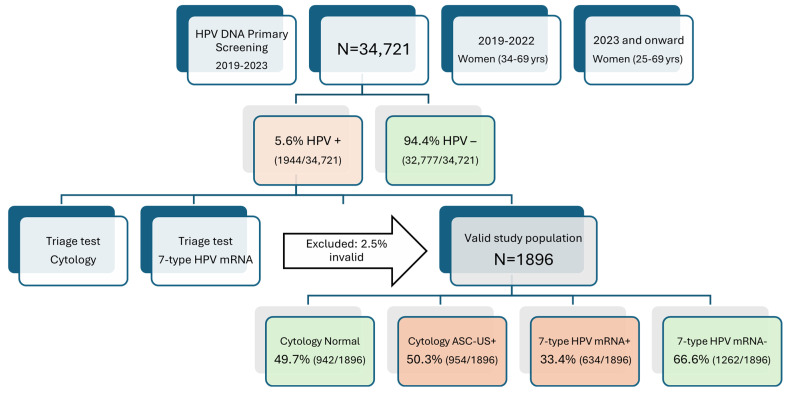
Flowchart of study population and triage cohort formation. Flow diagram outlining the inclusion of women from primary HPV DNA screening through triage and histological follow-up. Among 34,721 women screened between 2019 and 2023, 1944 (5.6%) tested positive for high-risk HPV DNA. After excluding samples with insufficient volume or invalid mRNA internal control, 1896 women were included in the analytic cohort. All underwent cytology and 7-type HPV mRNA testing, with follow-up for histologically confirmed CIN2+ through October 2024. Green-colored boxes indicate groups with lower CIN2+ risk, while red-colored boxes represent groups with elevated CIN2+ risk.

**Figure 2 pathogens-14-00749-f002:**
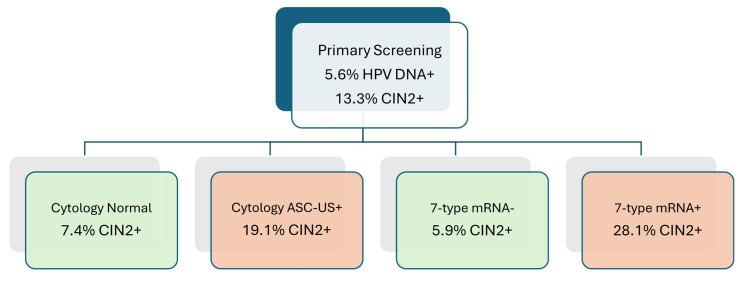
CIN2+ prevalence stratified by cytology and HPV mRNA triage results. CIN2+ prevalence among HPV DNA-positive women stratified by triage test result. Among women with abnormal cytology (≥ASC-US), 19.1% had CIN2+, compared to 7.4% with normal cytology. For the 7-type HPV mRNA test, CIN2+ was detected in 28.1% of mRNA-positive women and 5.9% of mRNA-negative women. These findings demonstrate improved risk discrimination with genotype-specific mRNA triage. Green-colored boxes represent groups with lower CIN2+ risk, while red-colored boxes represent groups with elevated CIN2+ risk.

**Figure 3 pathogens-14-00749-f003:**
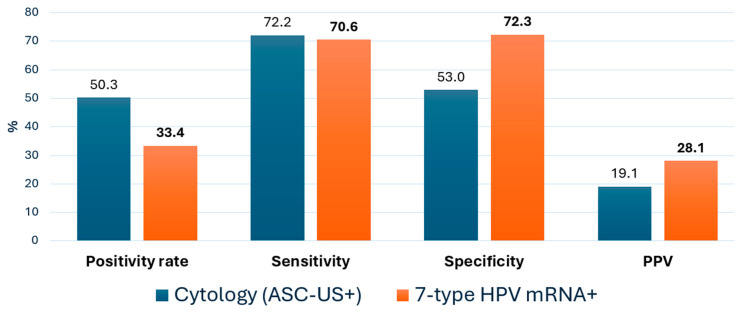
Comparative performance metrics for CIN2+ detection: Cytology vs. 7-Type HPV mRNA triage. Bar graph comparing test positivity rate, sensitivity, specificity and positive predictive value (PPV) for CIN2+ detection using cytology (≥ASC-US) versus the 7-type HPV E6/E7 mRNA assay. Values are based on the cohort of 1896 HPV DNA-positive women with histological follow-up.

**Figure 4 pathogens-14-00749-f004:**
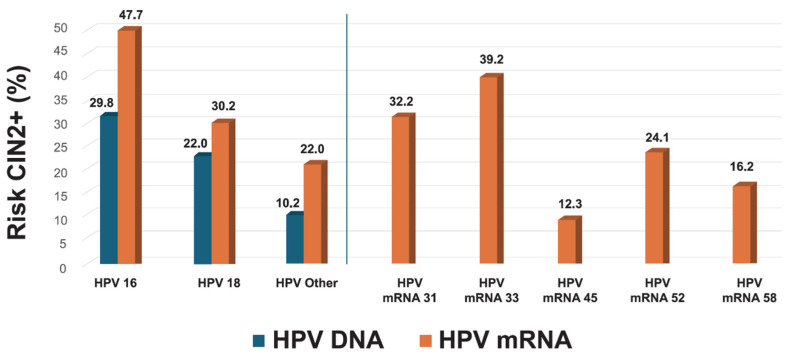
Genotype-specific risk stratification for CIN2+: Comparison of HPV DNA and mRNA testing. Positive predictive values (PPVs) for CIN2+ are shown by genotype, comparing HPV DNA and mRNA detection. The figure presents DNA-based PPVs for HPV16, HPV18, and a pooled group of 12 high-risk HPV types, alongside mRNA-based PPVs for HPV16, HPV18, and a pooled group of five non-16/18 genotypes (HPV31, 33, 45, 52, 58). Individual mRNA-based PPVs are also shown for each of the seven genotypes included in the assay. This analysis illustrates the superior risk discrimination provided by genotype-specific mRNA detection compared to DNA-based stratification.

**Figure 5 pathogens-14-00749-f005:**
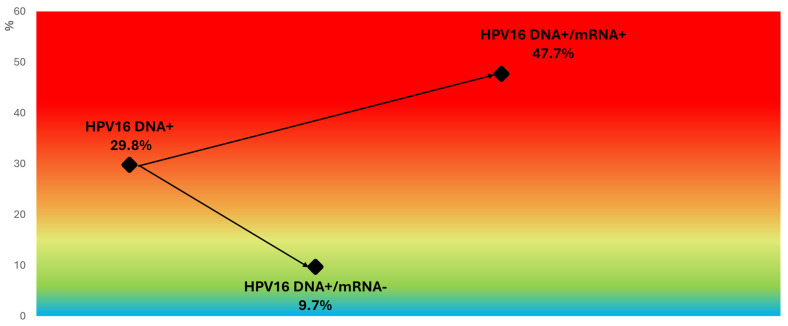
CIN2+ risk stratification in HPV16 DNA-positive women based on mRNA status. Color-gradient risk map illustrating CIN2+ prevalence among HPV16 DNA-positive women. Risk increases from 9.7% in mRNA-negative cases to 47.7% in mRNA-positive cases, demonstrating enhanced stratification achieved by detecting transcriptionally active HPV16. All categories refer to HPV16 DNA-positive women. Background gradient reflects increasing CIN2+ risk, ranging from blue (lower risk) to red (higher risk).

**Figure 6 pathogens-14-00749-f006:**
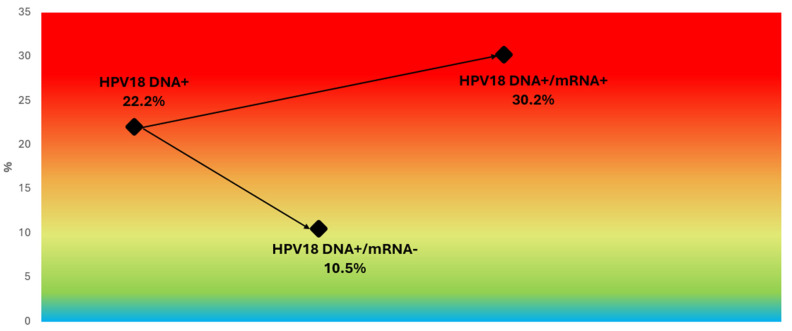
CIN2+ risk stratification in HPV18 DNA-positive women based on mRNA status. Color-gradient risk map showing CIN2+ prevalence among HPV18 DNA-positive women, stratified by mRNA status. Risk increased from 10.5% in mRNA-negative cases to 30.2% in mRNA-positive cases, highlighting the additional discriminatory value of transcriptional activity detection for HPV18 infections. All categories refer to HPV18 DNA-positive women. Background gradient reflects increasing CIN2+ risk, ranging from blue (lower risk) to red (higher risk).

**Figure 7 pathogens-14-00749-f007:**
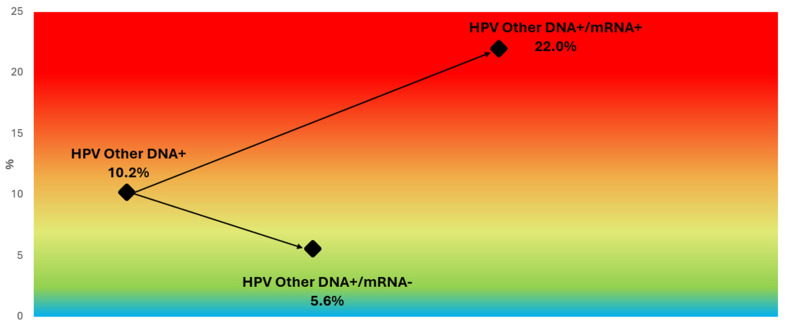
CIN2+ risk stratification in non-16/18 HPV DNA-positive women based on mRNA status. Color-gradient risk map showing CIN2+ prevalence among women positive for high-risk HPV DNA genotypes other than 16 and 18. Risk was 5.6% in mRNA-negative cases and 22.0% in mRNA-positive cases, demonstrating the ability of genotype-specific mRNA testing to refine risk estimates even within pooled non-16/18 infections. All categories refer to women who were HPV DNA-positive for at least one of the following genotypes: HPV31, 33, 45, 52, or 58. Background gradient reflects increasing CIN2+ risk, ranging from blue (lower risk) to red (higher risk).

**Table 1 pathogens-14-00749-t001:** Comparative diagnostic performance of cytology and 7-Type HPV mRNA triage for CIN2+ detection.

Metric Value	Cytology ≥ ASC-US % (95% CI)	7-Type HPV mRNA% (95% CI)	Δ (mRNA-Cyt) pp *
Sensitivity	72.2 (66.3–77.5)	70.6 (64.6–75.9)	−1.6
Specificity	53.0 (50.3–55.7)	72.3 (69.8–74.7)	+19.3
PPV	19.1 (16.4–22.0)	28.1 (24.9–31.6)	+9.0
NPV	93.0 (91.1–94.6)	94.6 (92.8–96.0)	+1.6

* Sensitivity, specificity, positive predictive value (PPV), and negative predictive value (NPV) for histologically confirmed CIN2+ were calculated for cytology (≥ASC-US) and the 7-type HPV mRNA test. Confidence intervals (95% CI) were calculated using the Clopper–Pearson method. Sensitivity was comparable between the two tests. The 7-type HPV mRNA test demonstrated substantially higher specificity and PPV compared with cytology. While absolute differences in specificity and NPV appear modest, they are clinically meaningful given the large sample size. Conversely, despite a clinically relevant increase in PPV with the mRNA test, the number of discordant CIN2+ cases was limited, indicating caution in interpreting statistical significance in paired analyses.

## Data Availability

The data presented in this study are available on request from the corresponding author. The data are not publicly available due to privacy and ethical restrictions in accordance with Norwegian data protection regulations.

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
