# Peer review of "Genotype-Specific HPV mRNA Triage Improves CIN2+ Detection Efficiency Compared to Cytology: A Population-Based Study of HPV DNA-Positive Women"

_pathogens, 2025, doi:10.3390/pathogens14080749_

Round 1
Reviewer 1 Report
Comments and Suggestions for Authors
Dear Authors,
I would like to commend you for your hard work and thorough research, which makes a significant contribution to the field of HPV-related issues. The manuscript is well-structured, clearly outlining the study group, treatments, analyses, and follow-up procedures. The results are presented with precision, and the discussion thoughtfully integrates the findings into the broader context of HPV research. The conclusion is well-supported by the data, demonstrating a understanding of the subject.
I recommend a minor revision. Please consider the following comments:
- Line 144-150: The current description of Figure 1, particularly the lower section presenting cytology and HPV mRNA triage results, may be confusing for readers. Please revise the text to clearly state that the 320 fewer referrals refer to women who were cytology-positive (≥ASC-US) but mRNA-negative, and therefore would not have been referred for colposcopy if mRNA triage had been used instead of cytology.
- Regarding Figure 5, please consider adding labels directly to the figure to clarify that all categories refer to HPV16 DNA-positive women. For example: “HPV 16+” should be changed to “HPV16 DNA+”; “HPV16+/mRNA16+” should be changed to “HPV16 DNA+/mRNA+”; and “HPV16+/mRNA16−” should be changed to “HPV16 DNA+/mRNA−”. Also, consider including a small legend or explanatory note below the figure to indicate that the background gradient reflects increasing CIN2+ risk (from blue to red).
- Please apply the same labeling and explanatory approach to Figures 6 and 7.
Author Response
Reviewer 1: General comments: I would like to commend you for your hard work and thorough research, which makes a significant contribution to the field of HPV-related issues. The manuscript is well-structured, clearly outlining the study group, treatments, analyses, and follow-up procedures. The results are presented with precision, and the discussion thoughtfully integrates the findings into the broader context of HPV research. The conclusion is well-supported by the data, demonstrating a understanding of the subject. Our response: We thank Reviewer 1 for the positive and encouraging comments. We greatly appreciate the recognition of our efforts and are pleased that the study design, data presentation, and discussion were found to be clear and of value to the field of HPV research. Comments 1: Line 144-150: The current description of Figure 1, particularly the lower section presenting cytology and HPV mRNA triage results, may be confusing for readers. Please revise the text to clearly state that the 320 fewer referrals refer to women who were cytology-positive (≥ASC-US) but mRNA-negative and therefore would not have been referred for colposcopy if mRNA triage had been used instead of cytology. Our response: Thank you for this helpful suggestion. We have revised the text in lines 144–150 to clarify that the 320 fewer referrals refer specifically to women who had abnormal cytology (≥ASC-US) but tested negative with the mRNA assay, and therefore would not have been referred for colposcopy if mRNA triage had been used instead of cytology. This clarification should help avoid confusion and better convey the clinical implication of reduced over-referral with mRNA testing. The updated text now reads: "Among the 1,896 HPV DNA-positive women, 50.3% (954/1,896) had abnormal cytology results (≥ASC-US), while 49.7% (942/1,896) were cytology-negative. In contrast, the 7-type HPV mRNA assay was positive in 33.4% (634/1,896) and negative in 66.6% (1,262/1,896) of cases. This reflects a 34% relative reduction in test positivity compared to cytology. Notably, 320 women had abnormal cytology but tested negative on the mRNA assay; these women would not have been referred for colposcopy if mRNA triage had been used instead of cytology, indicating improved specificity and reduced over-referral with the mRNA test (Figure 1)." Comments 2: Regarding Figure 5, please consider adding labels directly to the figure to clarify that all categories refer to HPV16 DNA-positive women. For example: “HPV 16+” should be changed to “HPV16 DNA+”; “HPV16+/mRNA16+” should be changed to “HPV16 DNA+/mRNA+”; and “HPV16+/mRNA16−” should be changed to “HPV16 DNA+/mRNA−”. Also, consider including a small legend or explanatory note below the figure to indicate that the background gradient reflects increasing CIN2+ risk (from blue to red). Please apply the same labeling and explanatory approach to Figures 6 and 7. Our response: Thank you for this clear and constructive comment. We have revised the labels in Figure 5 to explicitly state “HPV16 DNA+” in all categories, as suggested, to ensure clarity for readers. In addition, we have added a brief explanatory note below the figure indicating that the background gradient reflects increasing CIN2+ risk, with colors ranging from blue (lower risk) to red (higher risk). Similar adjustments have also been applied to Figures 6 and 7 to maintain consistency across all genotype-specific risk figures.Reviewer 2 Report
Comments and Suggestions for Authors
General Comments
The study addresses a clinically relevant question and presents precise data from an established population-based screening program. However, the manuscript essentially confirms existing knowledge rather than introducing novel insights, and several key areas, particularly statistical rigour, methodological transparency, and contextual discussion, require improvement to meet publication standards.
Specific Comments
- While the application of mRNA triage in a national screening program is of interest, the study does not offer significant innovation over previously published work, especially from the same author group. Please clearly articulate what is new in this study (e.g., scale, follow-up period, implementation outcome) and how it advances the current literature.
- Consider simplifying the title and indicating the assay name (PreTect™) and study type. Strengthen the introduction by identifying the evidence gap being addressed, including a brief comparison with other triage tools (e.g., dual-stain cytology, methylation markers) and ending with a concise, specific research objective.
- In the Methodology, provide more detail on inclusion/exclusion criteria, especially regarding prior treatment, immunosuppression, and follow-up completeness.
- Include a description of how histology was adjudicated (single vs. double-read, CIN2 vs. CIN3 distinction).
- Elaborate on assay technical performance: RNA degradation control, signal cutoffs, reproducibility.
- Clarify how confounders (e.g., age, vaccination) were addressed either through stratification or as a limitation.
- A power or precision calculation would strengthen the robustness of the diagnostic accuracy claims.
- In Results, include formal statistical comparisons (e.g., McNemar’s test) for sensitivity, specificity, PPV, and NPV differences between cytology and mRNA.
- Present separate data for CIN2 and CIN3+, or discuss their clinical relevance.
- If available, show time-to-diagnosis or follow-up outcomes for mRNA-negative women.
- Consider subgroup analyses by age or genotype groupings (e.g., HPV31/33/45)
- In the Discussion, expand the limitations section to acknowledge the Absence of vaccination stratification, Heavy reliance on previous in-house data and Lack of cost-effectiveness analysis or implementation feasibility
- Integrate findings from international trials (e.g., ATHENA, POBASCAM) to place results in a global context.
- Briefly address false negatives and the clinical safety of mRNA negative follow-up.
- Include recent literature on alternative triage methods and mRNA test performance in vaccinated cohorts.
I think it's a good idea to revise at this stage, with attention to statistical rigour, methodological transparency, and framing within the current literature.
Author Response
Reviewer 2:
General Comments
The study addresses a clinically relevant question and presents precise data from an established population-based screening program. However, the manuscript essentially confirms existing knowledge rather than introducing novel insights, and several key areas, particularly statistical rigour, methodological transparency, and contextual discussion, require improvement to meet publication standards.
Our response: We thank Reviewer 2 for recognizing the clinical relevance of our study and the quality of the population-based data presented. While we agree that our findings support and extend previous work, we believe the expanded cohort size, longer follow-up, and detailed genotype-specific analysis provide added value and strengthen the evidence for implementing HPV mRNA triage in routine screening. In response to the reviewer’s comments, we have made several revisions to improve methodological clarity, statistical transparency, and contextual discussion, as detailed below.
Specific Comments
Comments 1: While the application of mRNA triage in a national screening program is of interest, the study does not offer significant innovation over previously published work, especially from the same author group. Please clearly articulate what is new in this study (e.g., scale, follow-up period, implementation outcome) and how it advances the current literature.
Our response: Thank you for this important comment. We have revised the introduction to more clearly articulate the novel aspects of this study. Compared to our previous interim report (n = 962), the current analysis is based on a substantially expanded cohort (n = 1,896) and includes a full five-year implementation period (2019–2023) with extended histological follow-up through October 2024. We now provide detailed genotype-specific analyses of mRNA performance and its added value over DNA genotyping, which were not previously published. These updates allow for a more robust evaluation of diagnostic accuracy, colposcopy efficiency, and risk stratification by genotype. Collectively, this strengthens the evidence base for broader implementation of mRNA triage in routine screening. The revised paragraph in the introduction now reads:
“Building on prior data from the initial implementation phase, this study provides a substantially expanded analysis based on a doubled cohort size (1,896 women) and five years of real-world triage data. In addition to updated performance metrics with extended follow-up through October 2024, we provide detailed genotype-specific risk stratification analyses not previously reported. These results offer new evidence supporting the clinical utility of the 7-type HPV mRNA assay and its implementation feasibility in population-based cervical cancer screening.”
Comments 2: Consider simplifying the title and indicating the assay name (PreTect™) and study type.
Our response: Thank you for the suggestion. We agree that including the study type in the title could enhance clarity. However, we have chosen not to include the assay brand name (PreTect™), in line with journal conventions that discourage the use of proprietary or trademarked terms in titles. To address your comment, we propose the following revised title that maintains clarity and scientific neutrality while reflecting the nature of the study:
“Genotype-Specific HPV mRNA Triage Improves CIN2+ Detection Efficiency Compared to Cytology: A Population-Based Study of HPV DNA-Positive Women”
Comments 3: Strengthen the introduction by identifying the evidence gap being addressed, including a brief comparison with other triage tools (e.g., dual-stain cytology, methylation markers) and ending with a concise, specific research objective.
Our response: Thank you for this helpful suggestion. We have revised the introduction to more explicitly define the evidence gap and provide a brief comparison with other emerging triage methods such as dual-stain cytology and methylation markers. We also conclude the introduction with a clearer, specific research objective. The revised paragraph now reads:
“To address this, several triage strategies have been proposed for HPV DNA-positive women. Cytology remains the current standard, but its inherent subjectivity and limited reproducibility constrain its effectiveness. Partial genotyping offers some improvement by identifying high-risk types, yet it does not capture the transcriptional activity that drives disease progression. Additional molecular-based triage approaches, including dual-stained cytology (p16/Ki-67) and DNA methylation markers, have shown promise in improving objectivity and sensitivity. However, these methods often rely on cytological infrastructure, have variable specificity in real-world settings, and are not validated for self-collected samples—limiting their scalability in population-based programs. Thus, there remains a need for validated triage tools that combine high specificity with operational flexibility.”
“This study aims to evaluate the clinical performance of the genotype-specific 7-type HPV mRNA assay as a triage tool for HPV DNA-positive women in a large, real-world population-based screening program, comparing its diagnostic accuracy and risk stratification ability to cytology.”
Comments 4: In the Methodology, provide more detail on inclusion/exclusion criteria, especially regarding prior treatment, immunosuppression, and follow-up completeness.
Our response: Thank you for this important comment. We have now clarified the inclusion and exclusion criteria in the Materials and Methods section. As this study is based on a primary screening population within the national cervical cancer screening program in Norway, women with a history of cervical cancer, previous treatment for CIN2+ within the last 10 years, or ongoing follow-up of abnormal cytology or positive HPV tests were not included. Immunosuppressed women are managed outside the primary screening cohort and were therefore also excluded. Due to the nationwide call/recall system and use of unique national identification numbers, follow-up completeness is ensured both at the national and local level. This allows us to capture complete screening and outcome data, regardless of which laboratory or provider conducted the follow-up.
We have added the following sentence to Section 2.1:
“The study population consisted of women undergoing routine primary HPV screening. Women with a history of cervical cancer, prior treatment for CIN2+ within the last 10 years, ongoing follow-up for previous abnormal findings, or known immunosuppression were not included in the primary screening cohort. Completeness of follow-up was ensured through the national call/recall system and use of unique personal identifiers, which enable tracking of all cervical screening and follow-up data across laboratories and healthcare providers nationwide.”
Comments 5: Include a description of how histology was adjudicated (single vs. double-read, CIN2 vs. CIN3 distinction).
Our response: Thank you for this helpful comment. We have now clarified the histological evaluation process in the Materials and Methods section. All cervical biopsies with suspected high-grade lesions (CIN2+) were reviewed by two experienced pathologists. In diagnostically challenging cases, immunostaining with p16 was used to distinguish CIN1 from CIN2+. CIN grading followed WHO criteria. To support diagnostic consistency, we routinely use digital pathology with quality assurance supported by machine learning and EagleEye AI software. Final diagnostic decisions were made by the evaluating pathologist in case of discrepancy.
We have added the following sentences to Section 2.2:
“All high-grade cervical biopsies (CIN2+) were evaluated by two experienced pathologists. CIN grading was performed according to WHO criteria, with p16 immunostaining used in morphologically ambiguous cases to distinguish CIN1 from CIN2+. Digital pathology and machine learning-based quality assurance tools (including EagleEye AI) were used to support diagnostic consistency. In cases of discrepancy, the final diagnosis was determined by the evaluating pathologist.”
Comments 6: Elaborate on assay technical performance: RNA degradation control, signal cutoffs, reproducibility.
Our response: Thank you for this helpful comment. We have revised the 2.1.2 Triage Procedures section to include additional details regarding the technical performance of the PreTect HPV-Proofer`7 assay. Specifically, we have clarified the role of the intrinsic sample control (ISC) for monitoring RNA integrity, added a description of the signal-to-cutoff (S/CO) thresholds used for result interpretation, and noted that indeterminate results were retested per the assay protocol. Furthermore, we have stated that amplification is performed at 41 °C using NASBA technology, and that all procedures were carried out in accordance with the manufacturer’s instructions. While analytical validation was not the primary focus of this clinical performance study, we have also noted that the assay has undergone full analytical validation under CLIA standards and is currently offered as a laboratory-developed test (LDT) in a CAP-accredited clinical laboratory in the United States.
Comments 7: Clarify how confounders (e.g., age, vaccination) were addressed either through stratification or as a limitation.
Our response: Thank you for raising this point. We have clarified in the Discussion (Section 4.7) that HPV vaccination was not a significant confounding factor in this study. In Norway, primary HPV DNA screening was implemented in 2019 for women aged 34–69 years, a population largely unvaccinated. Although the national HPV vaccination program for 12-year-old girls began in 2009 with high uptake (70–90%), these cohorts only began entering the screening age group (≥25 years) in 2022 and were not included in the primary screening cohort until national guidelines were expanded to cover women aged 25–69 from July 2023. Therefore, only a small proportion of women in our study would have been vaccinated, and vaccination status is unlikely to have influenced the results. This clarification has been added to the limitations section to explain why stratification by vaccination status was not performed.
Comments 8: A power or precision calculation would strengthen the robustness of the diagnostic accuracy claims.
Our response: We appreciate this comment. However, as this is a retrospective quality-assurance study based on a population-based screening cohort, all eligible women who participated in the screening program during the study period were included. In this context, power calculations are not typically performed in advance, as the sample size is determined by the natural screening volume rather than by predefined inclusion targets.
Regarding precision, we have already presented 95% confidence intervals for all key performance metrics (sensitivity, specificity, PPV, and NPV), which reflect the statistical precision of our estimates. We hope this approach sufficiently conveys the robustness of our findings.
Comments 9: In Results, include formal statistical comparisons (e.g., McNemar’s test) for sensitivity, specificity, PPV, and NPV differences between cytology and mRNA.
Our response: Thank you for this constructive suggestion. We have now performed McNemar’s test to compare the sensitivity and specificity of cytology versus HPV mRNA triage for CIN2+ detection. The difference in sensitivity was not statistically significant (p = 0.13), whereas the improvement in specificity with the mRNA test was highly significant (p < 0.001). These p-values have been added to Section 3.4.
As PPV and NPV are prevalence-dependent and not directly paired at the individual level, McNemar’s test is not appropriate for those comparisons. Instead, we report exact 95% confidence intervals to reflect the precision of these estimates. We also note that the improved specificity of the mRNA test largely accounts for its higher PPV.
Comments 10: Present separate data for CIN2 and CIN3+, or discuss their clinical relevance.
Our response: Thank you for this important comment. In our study, we used CIN2+ as the primary clinical endpoint in line with national guidelines, where CIN2, CIN3, and AIS are all considered indications for treatment with LEEP, particularly in women aged ≥30 years. While CIN3+ is often regarded as a more robust and reproducible endpoint, the diagnostic criteria for CIN2 have improved with the use of p16 immunostaining, as recommended by the Lower Anogenital Squamous Terminology (LAST) project.
In Norway, only p16-positive CIN2 cases are classified as true CIN2 and warrant clinical intervention. Cases with ambiguous morphology that are p16-negative are downgraded to CIN1 or reactive changes. In this study, all CIN2+ diagnoses were confirmed by two independent pathologists, and ambiguous cases were adjudicated with p16 staining and supported by digital pathology tools, including EagleEye AI, to ensure diagnostic consistency. Therefore, we regard all CIN2+ lesions in our cohort as clinically meaningful and requiring treatment, consistent with national practice.
Moreover, our primary screening population consisted mainly of women aged 34–69 years, where the risk of CIN3+ is relatively low due to previous negative screening. As such, using CIN3+ as the sole outcome would have limited statistical power. Nonetheless, we agree that CIN3+ may serve as a more conservative benchmark in future evaluations, and we are planning extended follow-up and analyses with larger populations where CIN3+ will be included as a separate endpoint.
We have added the following clarification to the Discussion section:
“In Norway, only p16-positive CIN2 lesions are considered clinically significant and are classified as CIN2+. Ambiguous or p16-negative lesions are downgraded to CIN1 or reactive changes. All CIN2+ outcomes in this study were confirmed by two independent pathologists and, where needed, supported by p16 staining and digital pathology tools to ensure reproducibility and clinical relevance.”
Comments 11: If available, show time-to-diagnosis or follow-up outcomes for mRNA-negative women.
Our response: Thank you for this important suggestion. As this was a quality-assurance study, all clinical management and follow-up were based solely on HPV DNA and cytology results, in accordance with national guidelines. HPV mRNA testing was performed retrospectively and did not influence referral decisions or time to diagnosis. This clarification has been added to section 2.2 of the Materials and Methods.
Among mRNA-negative women, any delayed CIN2+ diagnoses were due to normal cytology and subsequent follow-up intervals recommended by the guidelines (e.g., repeat HPV testing after 12–24 months), not to false-negative mRNA results. Sensitivity for CIN2+ detection was comparable between cytology and the HPV mRNA test, while specificity was significantly higher for mRNA, supporting its ability to safely reduce unnecessary colposcopies.
Most mRNA-negative women with normal cytology had a negative HPV DNA test on follow-up and returned to routine screening. Women with persistent HPV DNA positivity were referred to colposcopy regardless of mRNA status. These patterns confirm that mRNA-negative results were associated with low risk and support the clinical safety of genotype-specific HPV mRNA testing as a triage tool.
Comments 12: Consider subgroup analyses by age or genotype groupings (e.g., HPV31/33/45)
Our response: Thank you for this valuable suggestion. We have included subgroup analyses by genotype in section 3.6 (Genotype-specific predictive values) and expanded on this in section 3.7 (Refining Risk Stratification in HPV16/18 DNA-Positive Women). These sections present positive predictive values (PPVs) for individual mRNA genotypes, including HPV31, 33, 45, 52, and 58, and demonstrate improved risk stratification compared to the corresponding DNA genotypes. Specifically, genotype-specific PPVs were highest for HPV33 (39.2%), HPV31 (32.2%), and HPV52 (24.1%), highlighting the discriminatory value of the 7-type mRNA assay.
While age-stratified analyses were not performed in the current study, we acknowledge that age may influence test performance, particularly in younger women where transient infections and CIN2 regression are more common. However, the majority of the cohort were women aged 34–69 years; only those screened in 2023 included women aged 25–33 years. Future studies should examine age-specific risk stratification in mRNA triage to further refine management recommendations.
Comments 13: In the Discussion, expand the limitations section to acknowledge the Absence of vaccination stratification.
Our response: Thank you for this important observation. We have revised the limitations section of the Discussion to explicitly acknowledge the absence of individual-level HPV vaccination data and the lack of vaccination-stratified analyses. While most women included in the study were born before the start of the national HPV vaccination program in 2009 and are likely unvaccinated, the inability to stratify by vaccination status remains a limitation, particularly as genotype distribution is expected to shift in younger, vaccinated cohorts. This point has now been clearly stated in the revised manuscript.
Comments 14: Heavy reliance on previous in-house data and Lack of cost-effectiveness analysis or implementation feasibility.
Our response: Thank you for highlighting this important point. While this study did not include a formal cost-effectiveness analysis, we agree that economic considerations and implementation feasibility are critical for evaluating triage strategies. Cytology-based triage is labor-intensive, subjective, and increasingly resource-limited, particularly in settings like Norway where most cervical samples are collected by general practitioners and access to gynecologists is constrained. The addition of genotype-specific HPV mRNA testing, although associated with a modest additional cost, has the potential to substantially reduce unnecessary colposcopies and biopsies by improving test specificity. This could translate to lower overall healthcare burden and more efficient use of specialist resources. We have added a statement in the Discussion to reflect these considerations and to encourage future studies on the cost-effectiveness and practical implementation of HPV mRNA triage.
Comments 15: Integrate findings from international trials (e.g., ATHENA, POBASCAM) to place results in a global context.
Our response: Thank you for this suggestion. We have included findings from the ATHENA trial to contextualize our results within the broader literature on molecular triage strategies. Specifically, the ATHENA sub-study on p16/Ki-67 dual-stained cytology is discussed in section 4.4 to highlight comparative performance. While the POBASCAM trial provided valuable evidence supporting HPV DNA testing in primary screening, it focused on cytology-based triage and did not assess alternative biomarkers such as HPV mRNA or dual-staining. Therefore, we considered it less directly relevant to our focus on mRNA-based triage. However, we acknowledge its contribution to shaping international screening policy and can reference it briefly for context if required.
Comments 16: Briefly address false negatives and the clinical safety of mRNA negative follow-up.
Our response: Thank you for highlighting this important point. In our study, there were no safety concerns associated with negative HPV mRNA results, as all women were managed and followed up based on HPV DNA and cytology findings according to national guidelines. Although no screening test is perfectly sensitive, the 7-type HPV mRNA assay demonstrated sensitivity for CIN2+ comparable to cytology.
In a future setting where mRNA triage may replace cytology, the high specificity of the mRNA assay could reduce unnecessary colposcopies without compromising safety—provided that HPV DNA-positive, mRNA-negative women continue to receive appropriate follow-up in line with national guidelines. These findings support the clinical safety of mRNA-negative results and their potential role in optimizing triage strategies.
We have updated Section 4.8 of the Discussion to incorporate this point.
Comments 17: Include recent literature on alternative triage methods and mRNA test performance in vaccinated cohorts.
Our response: Thank you for this suggestion. We conducted an updated literature search using Elicit and relevant databases to identify recent studies evaluating HPV mRNA test performance in vaccinated populations and comparisons with alternative triage methods such as dual-stained cytology (p16/Ki-67) and methylation markers.
To date, few studies have specifically assessed HPV mRNA test performance in vaccinated cohorts, likely due to the relatively recent entry of vaccinated women into screening age. Most published evidence, including our own, is still based on largely unvaccinated populations. We acknowledge this gap and have added this as a limitation in the discussion.
Regarding alternative triage strategies, we have already cited key studies, including the ATHENA trial (Wright et al., 2017), which evaluated dual-stained cytology, and others comparing molecular triage tools. A recent study by Giorgi Rossi et al. (2020) compared p16/Ki-67 dual staining and the APTIMA HPV mRNA test (14-type) for triage of HPV DNA-positive women. However, in that study, APTIMA was less specific than both cytology and dual-stained cytology, limiting its utility in triage. In contrast, our study focuses on a genotype-specific 7-type E6/E7 mRNA assay (PreTect), which has shown substantially higher specificity than cytology.
We have now included the following in the revised discussion (Section 4.4):
“While dual-stained cytology and methylation markers have shown promise as triage tools [27–30], some mRNA assays, such as the 14-type APTIMA test, have demonstrated lower specificity than cytology or dual-staining in head-to-head comparisons [Giorgi Rossi et al., 2020]. In contrast, the genotype-specific 7-type HPV mRNA assay evaluated in this study demonstrated significantly higher specificity than cytology, supporting its potential for reducing unnecessary colposcopy referrals.”
We also added the following to the limitations:
“Few studies have evaluated the performance of HPV mRNA triage in vaccinated populations. As vaccination coverage continues to rise and genotype prevalence shifts, future research will be essential to assess the effectiveness and clinical utility of mRNA-based triage strategies in vaccinated cohorts.”
Round 2
Reviewer 2 Report
Comments and Suggestions for Authors
I appreciate the thorough and well-structured revisions made to the manuscript. The updated version significantly improves methodological clarity, statistical transparency, and contextual relevance. Overall, the revised manuscript presents clinically meaningful findings that strengthen the evidence base for mRNA-based triage strategies in cervical cancer screening. I recommend the manuscript for publication.